

# DGPathinter: a novel model for identifying driver genes via knowledge-driven matrix factorization with prior knowledge from interactome and pathways

Jianing Xi[1,*], Minghui Wang[1,2,*] and Ao Li[1,2]

[1] School of Information Science and Technology, University of Science and Technology of China, Hefei, China
[2] Centers for Biomedical Engineering, University of Science and Technology of China, Hefei, China
* These authors contributed equally to this work.

## ABSTRACT

Cataloging mutated driver genes that confer a selective growth advantage for tumor cells from sporadic passenger mutations is a critical problem in cancer genomic research. Previous studies have reported that some driver genes are not highly frequently mutated and cannot be tested as statistically significant, which complicates the identification of driver genes. To address this issue, some existing approaches incorporate prior knowledge from an interactome to detect driver genes which may be dysregulated by interaction network context. However, altered operations of many pathways in cancer progression have been frequently observed, and prior knowledge from pathways is not exploited in the driver gene identification task. In this paper, we introduce a driver gene prioritization method called driver gene identification through pathway and interactome information (DGPathinter), which is based on knowledge-based matrix factorization model with prior knowledge from both interactome and pathways incorporated. When DGPathinter is applied on somatic mutation datasets of three types of cancers and evaluated by known driver genes, the prioritizing performances of DGPathinter are better than the existing interactome driven methods. The top ranked genes detected by DGPathinter are also significantly enriched for known driver genes. Moreover, most of the top ranked scored pathways given by DGPathinter are also cancer progression-associated pathways. These results suggest that DGPathinter is a useful tool to identify potential driver genes.

# INTRODUCTION

In the last decade, studies based on advanced DNA sequencing technologies have highlighted the fact that the development and progression of cancer hinges on somatic abnormalities of DNA (*Hudson et al., 2010*; *Vogelstein et al., 2013*; *Raphael et al., 2014*). Despite a small number of driver genes conferring a selective growth advantage for tumor cells, a considerable number of somatic mutations are sporadic passenger mutations that have no impact on cancer process (*Sjöblom et al., 2006*; *Youn & Simon, 2011*;

Corresponding author
Ao Li, aoli@ustc.edu.cn

*Dees et al., 2012*; *Lawrence et al., 2013*; *Hua et al., 2013*; *Cho et al., 2016*). For this reason, distinguishing driver genes from genes with passenger mutations is a critical challenge for understanding genetic basis for cancer. At the same time, somatic mutations of genes in tumor samples can be efficiently detected by next generation sequencing technology (*Schuster, 2007*; *Xiong et al., 2011*; *Zhao et al., 2013*), and enormous accumulated datasets of cancer genomic alterations have been provided by studies such as the cancer genome atlas (TCGA) (*Weinstein et al., 2013*) and the International Cancer Genome Consortium (ICGC) (*Hudson et al., 2010*). These large-scale datasets of cancer genomics offer us an unprecedented opportunity to discover driver genes from the somatic mutation profiles of tumor samples (*Kandoth et al., 2013*; *Lawrence et al., 2013*, *2014*; *Tamborero et al., 2013*).

To address the driver and passenger problem, many efforts have been undertaken to catalogue genes by comparing the mutation frequencies of the tested genes with the background mutation rates (BMRs) through statistical analysis (*Dees et al., 2012*; *Lawrence et al., 2013*; *Hua et al., 2013*; *Sjöblom et al., 2006*; *Youn & Simon, 2011*). For example, a previous study has been adopted to identify genes with mutational significance by using a per-gene BMR (*Dees et al., 2012*), and another research study on driver genes is based on utilizing information of coverage and other genomic features such as DNA replication time to estimate the BMRs of genes (*Lawrence et al., 2013*). Furthermore, Bayesian approaches are also applied to estimate BMRs in detecting driver genes (*Hua et al., 2013*). In addition, some other studies are proposed to determine driver genes through the cancer mutation prevalence scores of genes in tumor samples (*Sjöblom et al., 2006*) or the predicted impact on protein function and the mutational recurrence of genes (*Youn & Simon, 2011*). Through these mutation frequency-based approaches, a number of statistically significantly potential driver genes have been identified (*Dees et al., 2012*; *Lawrence et al., 2013*; *An et al., 2014*).

Nevertheless, although some driver genes are mutated at high frequencies among tumor samples, previous studies have reported that some driver genes are mutated at low frequencies, and the mutation frequencies of these genes are too low to be tested as statistically significant (*Vandin, Upfal & Raphael, 2011*; *Leiserson et al., 2014*; *Raphael et al., 2014*). A prevalent assumption to explain the long tail phenomenon is that genes usually interact with other genes, and some genes with no mutation can be perturbed by their interacting neighbors (*Vandin, Upfal & Raphael, 2011*; *Leiserson et al., 2014*; *Raphael et al., 2014*; *Cho et al., 2016*). Based on this assumption, many studies for driver gene identification have been proposed by incorporating interactome information as prior knowledge (*Vandin, Upfal & Raphael, 2011*; *Leiserson et al., 2014*; *Raphael et al., 2014*; *Hofree et al., 2013*; *Bashashati et al., 2012*; *Cho et al., 2016*). The interactome information is employed as gene interaction network obtained from databases including iRefIndex (*Razick, Magklaras & Donaldson, 2008*), STRING (*Szklarczyk et al., 2011*) and others (*Prasad et al., 2009*; *Lee et al., 2011*; *Das & Yu, 2012*; *Khurana et al., 2013*). For example, HotNet and HotNet2 use the idea of heat-diffusion and propagate the mutation frequency scores of genes through the network, and calculate the significance scores of genes to identify potential driver genes (*Vandin, Upfal & Raphael, 2011*; *Leiserson et al., 2014*). NBS is an integrated method that propagates the mutations through the

interaction network for each tumor sample as preprocessing, and uses matrix factorization to obtain mutation-based subtypes and the mutation profiles of each subtypes (*Hofree et al., 2013*), where the mutation profiles can be utilized to prioritize driver genes (*Hofree et al., 2013*; *Shi, Gao & Wang, 2016*). Instead of using network propagation, MUFFINN prioritizes the genes by the mutational impact of their direct neighbors in the network context (*Cho et al., 2016*). In addition, interaction network information has also been used to predict patient specific driver genes, which helps the personalized analysis (*Hou & Ma, 2014*; *Jia & Zhao, 2014*; *Bertrand et al., 2015*). Through the network-based approaches, many novel potential driver genes have been discovered, which greatly complements the understanding of cancer driver genes (*Leiserson et al., 2014*; *Raphael et al., 2014*; *Cho et al., 2016*).

However, knowledge from pathways is not exploited in the aforementioned driver gene identification approaches. Since the operation of many pathways has been frequently reported to be altered in cancer progression (*Parsons et al., 2008*; *Cancer Genome Atlas Research Network, 2008*; *Vaske et al., 2010*), the knowledge from pathways is also important for understanding the roles of genes in cancer and thus can conduct the identification of cancer driver genes. Notably, some studies have cataloged pathway knowledge into publicly available databases, such as KEGG (*Ogata et al., 1999*), reactome (*Joshi-Tope et al., 2005*) and BioCarta (*Nishimura, 2001*), which have also been used to detect the perturbed pathways involved in the tested tumor samples in some previous efforts (*Subramanian et al., 2005*; *Ng et al., 2012*; *Li et al., 2016*; *Ma et al., 2016*). Although the pathway information is used in these approaches, they are not designed to identify potential driver genes. Meanwhile, the aforementioned driver gene detecting methods only use interactome information and not the pathway information. Consequently, the already available knowledge from pathways remains an underexploited resource in the identification of potential driver genes, and there is a lack of an approach that can effectively integrate information from both interactome information and pathways as prior knowledge.

In this article, we introduce driver gene identification through pathway and interactome information (DGPathinter), to discover potential driver genes from mutation data through a knowledge-based matrix factorization framework, where prior knowledge from pathways and interaction network is efficiently integrated. By maximizing the correlation between the relations of mutation scores of genes and the pathway scores (*Chen & Zhang, 2016*), we can identify potential driver genes driven by prior knowledge from pathways. At the same time, we also use a graph Laplacian technique to adopt information from an interaction network in the identification of driver genes (*Xie, Wang & Tao, 2011*). In addition, we use the framework of matrix factorization to integrate the information of mutation profiles, interactome and pathways, which is capable of factorizing the gene mutation scores from different sets of tumor samples and helps DGPathinter to address tumor sample heterogeneity issue (*Lee et al., 2010*; *Sill et al., 2011*; *Zhou et al., 2014*; *Xi & Li, 2016*). Compared with our previous approach (*Xi, Li & Wang, 2017*), DGPathinter is a revised computational model with additional prior information incorporated. Although both DGPathinter and the previous approach

(*Xi, Li & Wang, 2017*) utilize matrix factorization framework and network information, DGPathinter further considers the prior information of the pathway. In addition to driver gene identification, DGPathinter can provide highly scored pathways for the investigated tumor samples, while our previous approach could not. When we apply DGPathinter and three existing interactome driven methods on three TCGA cancer datasets, the detection results of DGPathinter outperform those of the competing methods. The top ranked genes detected by DGPathinter are also highly enriched for known driver genes. We further investigate the top ranked scored pathways yielded by DGPathinter, demonstrating that most of these pathways are also associated with cancer progressions. The remainder of the paper is organized as follow: "Materials and Methods" introduces the rationales and detailed techniques of our method DGPathinter. In "Results", we apply our method on three cancer datasets and evaluate DGPathinter with the three existing methods through known driver genes. Finally, we discuss our future work and make a brief conclusion in "Discussion". The code of DGPathinter can be freely accessed at https://github.com/USTC-HIlab/DGPathinter.

## MATERIALS AND METHODS

### Somatic mutation datasets

For the somatic mutation data of cancers, we focus on three types of cancers from TCGA datasets, which include 507 tumors samples from breast invasive carcinoma (BRCA) (*Cancer Genome Atlas Network, 2012*), 83 tumor samples from glioblastoma multiforme (GBM) (*Cancer Genome Atlas Research Network, 2008*) and 401 tumor samples from thyroid carcinoma (THCA) (*Cancer Genome Atlas Research Network, 2014*). The somatic mutation data are downloaded from cBioPortal database (*Gao et al., 2013*). The somatic mutation data are formed as a binary matrix (sample × gene) $X_{n \times p}$ (*Bashashati et al., 2012*; *Hofree et al., 2013*; *Kim, Sael & Yu, 2015*), where $n$ is the number of samples and $p$ is the number of the tested genes. An entry of the matrix being 1 denotes a mutation occurs in the respective gene and tumor sample, when compared with the germline (*Bashashati et al., 2012*; *Hofree et al., 2013*; *Kim, Sael & Yu, 2015*). The network information used in this study is iRefIndex (*Razick, Magklaras & Donaldson, 2008*), a highly curated interaction network containing 12,129 nodes (genes) and 91,809 edges (interactions). For the pathway information, we follow previous studies (*Park et al., 2015*) and use the curated pathways from three databases, KEGG (*Ogata et al., 1999*), reactome (*Joshi-Tope et al., 2005*) and BioCarta (*Nishimura, 2001*), which are also downloaded from the previous study (*Park et al., 2015*).

### Model of knowledge-driven matrix factorization

To efficiently identify potential driver genes from somatic mutation data, we use a knowledge-driven matrix factorization framework, which can successfully integrated information from pathways and interaction networks. A brief overview of DGFathinter is illustrated in Fig. 1. Since many matrix factorization-based methods have been used for detecting abnormal genes from heterogeneous tumor samples (*Lee et al., 2010*; *Sill et al., 2011*; *Zhou et al., 2013*, *2014*; *Xi & Li, 2016*; *Xi, Li & Wang, 2017*), we introduce matrix

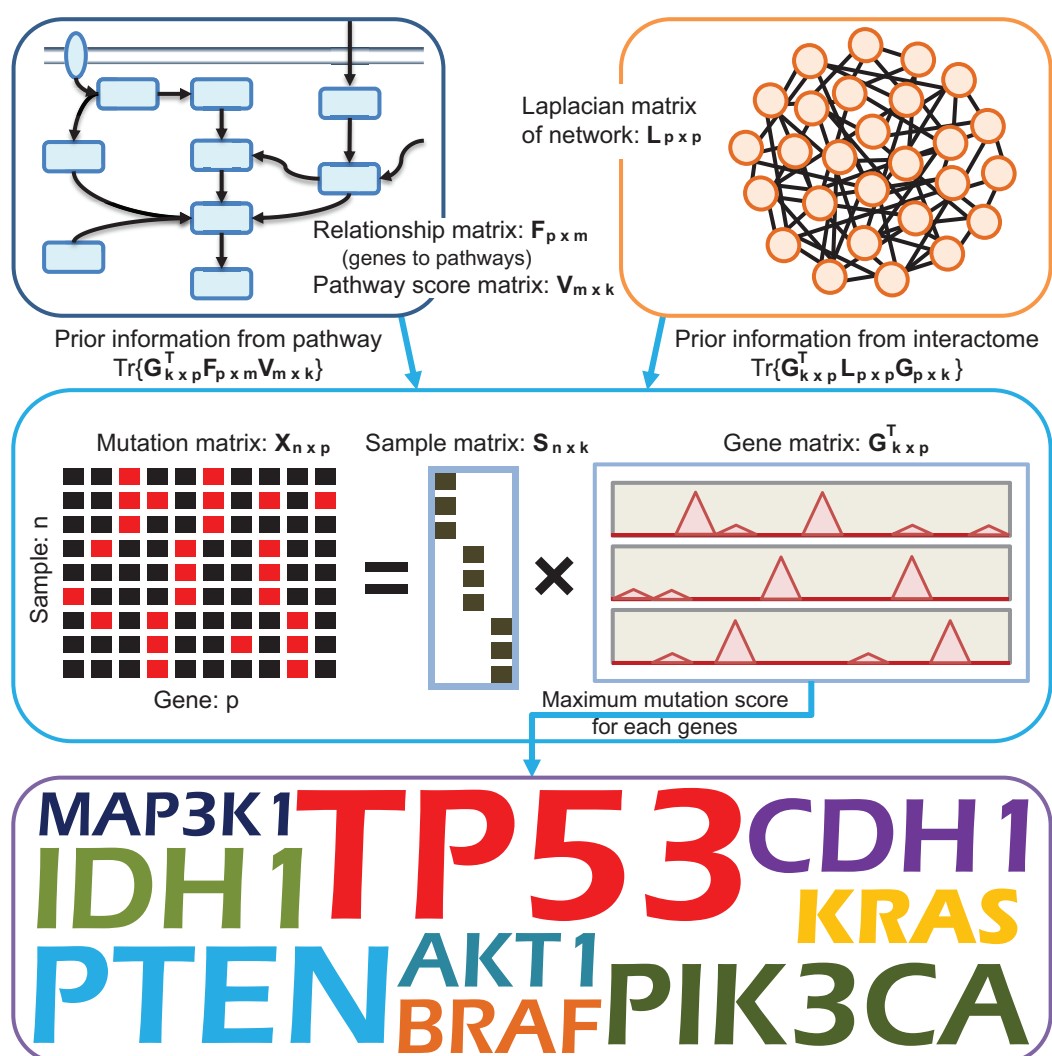

**Figure 1 A schematic diagram providing an overview of DGPathinter.** In DGPathinter, we utilize prior knowledge from pathways and interactome information in our model. The two types of prior knowledge are integrated via a knowledge-driven matrix factorization framework. This matrix factorization framework also decompose the somatic mutation matrix as the multiplication of two low rank matrices $S_{n \times k} = (s_1, \ldots, s_k)$ and $G^T_{k \times p} = (g_1, \ldots, g_k)^T$, which is equivalent to the summation of $k$ rank-one layers $\sum_{i=1}^{k} (s_i g_i^T)$. The matrix $S$ is a binary matrix, of which the entries represent to the assignments of the samples to the rank-one layers. The entries of the matrix $G^T$ denote the gene mutation scores for the samples in the rank-one layers. To integrate the pathway information into the analysis workflow, we project the gene scores in the matrix $G$ onto their related pathways and maximize the covariance between the projection scores and pathway scores $-\text{Tr}\{G^T FV\}$, where the bipartite matrix $F_{p \times m}$ represents the relationships of the genes and the pathways, and the entries of the non-negative pathway score matrix $V_{m \times k}$ represent the scores of the respective pathways and rank-one layers. Meanwhile, to incorporate interactome information from an interaction network, we introduce a graph Laplacian regularization term $\text{Tr}\{G^T LG\}$ on the matrix $G$, where the matrix $L_{p \times p}$ is the Laplacian matrix of the interaction network. For each gene, we choose the maximal gene mutation scores among the $k$ rank-one layers from the matrix $G$ and prioritize the driver genes. The top ranked genes are regarded as potential driver genes for further evaluations.

factorization framework into DGPathinter. The matrix factorization-based methods factorize the data matrix as the multiplication of a low-rank sample matrix and a low-rank gene matrix (*Lee et al., 2010*; *Sill et al., 2011*; *Zhou et al., 2013*, *2014*; *Xi & Li, 2016*; *Xi, Li & Wang, 2017*), where the entries of the sample matrix indicate the assignments of different samples to different subsets and the entries of the gene matrix indicate the scores of the abnormal genes in the related subsets of samples. In our previous study (*Xi, Li & Wang, 2017*), the matrix factorization framework has been shown to be an appropriate framework for the task of detecting driver genes from mutation data of heterogeneous cancers. Here we denote the matrix $G_{k \times p} = (g_1, \ldots, g_k)$ as the gene matrix and the binary matrix $S_{n \times k} = (s_1, \ldots, s_k)$ as the sample matrix, and use their multiplication $SG^T$ to approximate the mutation matrix $X$. The entries of $G$ represent the mutation scores of the tested genes related to the set of tumor samples indicated by the sample matrix $S$, which represent the assignments of the tested sample in different sample sets. Here the number $k$ is the rank of reconstruction matrix of the mutation matrix $X$. Due to the constraint that the entries of matrix $S$ are binary value, we use Boolean constraint on matrix $S$ (*Malioutov & Malyutov, 2012*), i.e., $S \circ (S - J) = 0$, where the operator $\circ$ indicates Hadamard product of two matrix, and the matrix $J$ denotes an $n \times k$ matrix with all the entries being 1. The fitting problem of the multiplication of the two matrices $S$ and $G$ and the mutation matrix $X$ can be formulate as $X \simeq SG^T + \varepsilon$, where the $\varepsilon$ is the residual matrix between the matrix $X$ and the multiplication $SG^T$, and matrix $S$ is subject to the equality restriction $S \circ (S - J) = 0$. By estimating the mutation scores of the tested genes in the matrix $G$ from information of somatic mutation data, pathways and interaction network, we can identify the potential driver genes by ranking their mutation scores. The strategies of incorporating pathway and network information are present below.

To make the driver gene prioritization procedure in our model driven by prior knowledge from pathways, we introduce a non-negative matrix $V_{m \times k}$ as the pathway score matrix. The row number of the matrix $m$ is total number of pathways used in our model. The column vectors in the matrix $V = (v_1, \ldots, v_k)$ represent the scores of the pathways, and a higher score of a pathway indicates a larger potential that the pathway is dysregulated in the related set of tumor samples. To incorporate pathway information into gene scores for different sets of samples, we project the gene scores onto their related pathways and maximize the covariance between the projection scores and pathway scores as $R_C = -\sum_{j=1}^{k} \text{Cov}(Fg_j, v_j) = -\text{Tr}\{G^T F^T V\}$ (*Chen & Zhang, 2016*), where the matrix $V$ is subject to the inequality restriction $V \geq 0$. Here the matrix $F_{m \times p}$ represents the relationships of the tested genes and their related pathways. The entry $F_{ij}$ equaling 1 denotes that the $j$th gene belongs to the $i$th pathway. In addition, to avoid an overfitting problem, we also use Frobenius norm-based regularization on the pathway scores $V$ as $R_V = \|V\|_F^2$ (*Pan et al., 2008*). Furthermore, to integrate interaction network information into our model, we utilize Laplacian regularization to encourage the smoothness between the scores of the interacted genes (*Xie, Wang & Tao, 2011*). The regularization term is formulated as $R_L = \text{Tr}\{G^T L G\}$, where the matrix $L = D - A$ is the Laplacian matrix of

interaction network, the matrix $A$ is the adjacency matrix, and the matrix $D$ is its corresponding degree matrix.

Consequently, we estimate the mutation scores of the tested genes in gene matrix $G$, pathway score matrix $V$ and sample indicator matrix $S$ by optimizing an optimization function of a knowledge-driven model with integrated data fitting and regularization terms formulated as:

$$\min_{S,G,V} \frac{1}{2}\left\|X - SG^{\mathrm{T}}\right\|_F^2 - \lambda_C \mathrm{Tr}\{G^{\mathrm{T}}F^{\mathrm{T}}V\} + \frac{1}{2}\lambda_{\mathrm{L}}\mathrm{Tr}\{G^{\mathrm{T}}LG\} + \frac{1}{2}\lambda_{\mathrm{V}}\|V\|_{\mathrm{F}}^2 \tag{1}$$
$$\mathrm{s.t.}\, S \circ (S - J) = 0, V \geq 0$$

where $\lambda_C$, $\lambda_L$ and $\lambda_V$ are used to balance the data fitting, the coherence gene scores and pathway scores according to their relations, the smoothness between scores of interacted genes and the regularization term of pathway scores. The tuning parameters $\lambda_C$, $\lambda_V$ and $\lambda_L$ are empirically set to 0.01, 0.01 and 0.1, respectively. We have also investigated the results of DGPathinter when the three parameters changes. For the three parameters, we can see that the detection results of the top 100 genes show little variance when the tuning parameters are changed (Figs. S1–S3), demonstrating the robustness of our model with respect to these parameters. In Fig. 1, we illustrate an overview of DGPathinter through schematic diagram.

## Optimization of knowledge-driven matrix factorization

Due to the equivalence between the matrix multiplication $SG^{\mathrm{T}}$ and the summation of multiple rank-one layers $\sum_{i=1}^{k} s_i g_i^{T}$, we incorporate a layer-by-layer procedure to solve the optimization problem iteratively (*Lee et al., 2010*; *Sill et al., 2011*; *Xi & Li, 2016*). Note that the first layer is the best rank-one estimation of the data matrix. We estimate the first layer by minimizing the following objective function

$$\min_{s_1,g_1,v_1} \frac{1}{2}\left\|X - s_1 g_1^{\mathrm{T}}\right\|_F^2 - \lambda_C \mathrm{Tr}\{g_1^{\mathrm{T}}F^{\mathrm{T}}v_1\} + \frac{1}{2}\lambda_{\mathrm{L}}\mathrm{Tr}\{g_1^{\mathrm{T}}Lg_1\} + \frac{1}{2}\lambda_{\mathrm{V}}\|v_1\|_F^2 \tag{2}$$
$$\mathrm{s.t.}\, s_1 \circ (s_1 - I_{\mathrm{n}}) = 0, v_1 \geq 0,$$

where $s_1$, $g_1$ and $v_1$ are the first column vectors of matrices $S$, $G$ and $V$ respectively, and $1_{n \times 1}$ indicate a vector with all coefficients being 1. The $v_1^{T} v_1$ is the inner product of vector $v_1$, which is equivalent to squared Frobenius norm of the vector.

We then apply an alternatively strategy to estimate the three vectors $s_1$, $g_1$ and $v_1$ in Eq. (2). When the other two vectors $v_1$ and $s_1$ are fixed, the minimization problem for the mutation score vector $g_1$ can be reformulated as below:

$$\min_{g_1} \frac{1}{2}\|s_1\|_2^2 g_1^{\mathrm{T}}g_1 - (X^{\mathrm{T}}s_1)g_1 - \lambda_C(F^{\mathrm{T}}v_1)^{\mathrm{T}}g_1 + \frac{1}{2}\lambda_L g_1^{\mathrm{T}}Lg_1. \tag{3}$$

Through Karush–Kuhn–Tucker (KKT) conditions, the mutation score vector $g_1$ can be estimated as

$$g_1 \leftarrow \left(\|s_1\|_2^2 I_p + \lambda_L L\right)^{-1}\left(X^T s_1 + \lambda_C F^T v_1\right),\tag{4}$$

where $I_p$ is a $p \times p$ identity matrix.

Likewise, the optimization function to solve the pathway score vector $v_1$ in optimization problem in Eq. (2) is formulated as

$$\min_{v_1} \frac{1}{2}\lambda_V v_1^T v_1 - \lambda_C (F g_1)^T v_1$$
$$\text{s.t.} v_1 \geq 0,\tag{5}$$

which is a non-negative quadratic programming problem. The estimation of the vector $v_1$ in Eq. (5) can be calculated as:

$$v_1 \leftarrow \{(\lambda_C/\lambda_V)F g_1\}^+,\tag{6}$$

where $\{\cdot\}^+$ is an operator which replace the negative coefficients of the input vector with zeros.

For sample indicator vector $s_1$, the optimization function of Eq. (2) is formulated as a Boolean constraint problem

$$\min_{g_1} \frac{1}{2}\|g_1\|_2^2 s_1^T s_1 - (X g_1)s_1$$
$$\text{s.t.} s_1 \circ (s_1 - I_n) = 0.\tag{7}$$

Through KKT conditions, the problem in Eq. (7) can be solved as:

$$s_1 \leftarrow I_{[0,+\infty)}\left(X g_1 - \tfrac{1}{2}\|g_1\|_2^2\right)\tag{8}$$

where $I_\Phi(z)$ is indicator function, of which the coefficients of the output vector are assigned to 1 when the corresponding coefficients of input vector $z$ belongs to the set $\Phi$, and 0 otherwise. Consequently, the minimizing function is optimized by alternatively estimating the three vectors $g_1$, $v_1$ and $s_1$ in Eqs. (4), (6) and (8) until convergence (Pseudo-code in Table 1).

After convergence, the first rank-one layer $s_1 g_1^T$ from the mutation matrix $X$, along with the related pathway score vector $v_1$, are obtained. Since the cancer data may display heterogeneity, it is not sufficient to utilize only one layer to fit the mutation data matrix. Subsequently, we apply the one layer estimation strategy aforementioned on the remaining samples to obtain the next layer. When the mutation matrix is factorized iteratively until no sample remains, we can obtain the rank number $k$ automatically (*Lee et al., 2010*; *Sill et al., 2011*; *Xi & Li, 2016*). The multiple layers estimation yielded by our model can effectively incorporate information from the mutation matrix, the interaction network and pathways.

## Experimental design and evaluation

For the driver gene prioritization of our approach, we select the maximum entries of each row of gene matrix $G$ as the score of the tested genes to be potential driver genes, which represent the intensities of the mutation of tested genes among different sets of

**Table 1 Pseudo-code of the first rank-one layer estimation of DGPathinter.**

**Algorithm 1** DGPathinter: iterative estimation of the first rank-one layer

| | |
|---|---|
| **Input:** | soamtic mutation matrix $X_{n \times p}$; pathway by gene bipartite matrix $F_{m \times p}$; graph Laplacian matrix of interaction network $L_{p \times p}$. |
| **Output:** | sample indicator vector $s_1$ ($n \times 1$); gene score vector $g_1$ ($p \times 1$); pathway score vector $v_1$ ($m \times 1$). |

1:     set $\lambda_C \leftarrow 0.01$, $\lambda_V \leftarrow 0.01$, $\lambda_L \leftarrow 0.1$ and $t \leftarrow 0$

2:     $s_1^{(0)} \leftarrow 1_{n \times 1}$, $v_1^{(0)} \leftarrow 0_{m \times 1}$ and $g_1^{(0)} \leftarrow \left(nI_p + \lambda_L L\right)^{-1}\left(X^{\mathrm{T}}s_1^{(0)} + \lambda_C F^{\mathrm{T}}v_1^{(0)}\right)$

3:     **repeat**

4:        $v_1^{(t+1)} \leftarrow \{(\lambda_C/\lambda_V)Fg_1^{(t)}\}^+$

5:        $g_1^{(t+1)} \leftarrow \left(\left\|s_1^{(t)}\right\|_2^2 I_p + \lambda_L L\right)^{-1}\left(X^{\mathrm{T}}s_1^{(t)} + \lambda_C F^{\mathrm{T}}v_1^{(t+1)}\right)$

6:        $s_1^{(t+1)} \leftarrow I_{[0,+\infty)}\left(Xg_1^{(t+1)} - \frac{1}{2}\left\|g_1^{(t+1)}\right\|_2^2\right)$

7:        $t \leftarrow t + 1$

8:     **until** Convergence

9:     **return** $v_1 \leftarrow v_1^{(\infty)}$, $g_1 \leftarrow g_1^{(\infty)}$ and $s_1 \leftarrow s_1^{(\infty)}$

Notes:
$1_{n \times 1}$ is an $n \times 1$ vector with all coefficients being 1;
$0_{m \times 1}$ is an $m \times 1$ vector with all coefficients being 0.

tumor samples. We then prioritize the investigated genes according to their mutation scores and select the top ranked genes as potential driver genes. Due to the lack of gold standard for driver genes that are generally accepted, we further evaluate the detected genes via a list of known benchmarking cancer genes from a highly curated database, the Network of Cancer Genes (NCG4.0) (*An et al., 2014*). The NCG4.0 gene list contains both experimentally supported cancer genes from the Cancer Gene Census (CGC) (*Futreal et al., 2004*) and statistical inferred candidate genes from previous studies (*An et al., 2014*). The cancer specific genes from the two benchmarking gene lists are used to assess the prioritizing results of the investigated methods.

By using these benchmarking genes as ground truth genes in the evaluation studies, we firstly compute the precisions and recalls under different rank thresholds and draw precision–recall curves of the competing methods, where a curve closer to the top and right indicates a better performance (*Wu, Hajirasouliha & Raphael, 2014*; *Yang et al., 2017*). The precision is calculated as the fraction of selected genes that are also benchmarking genes, and the recall is computed as the fraction of benchmarking genes that are selected by the rank threshold. Next, we calculate the average rank of known genes in the prioritization results to comprehensively assess the prioritization performance, which is a traditional metric for evaluating the performance of retrieval (*Ma & Zhang, 1998*; *Gargi & Kasturi, 1999*; *Müller et al., 2001*). Furthermore, we select the top 100 genes from the results of the competing methods, and compare the proportions of known driver genes detected by the competing methods. Fisher's exact test is also applied on the results, which can evaluate whether the selected genes are significantly enriched for known driver genes by $p$ values of the test. In addition, for the highly scored pathways given by our

approach, we also investigate whether these pathways are correlated with cancer progressions.

# RESULTS

## Driver gene identification

To evaluate the identification performance of DGPathinter, we compare our model with three existing methods, HotNet2 (*Leiserson et al., 2014*), NBS (*Hofree et al., 2013*) and MUFFINN (*Cho et al., 2016*), on datasets of three types of cancers: BRCA, GBM and THCA. In the performance evaluation, HotNet2, NBS and MUFFINN are set to their default parameters. For MUFFINN, there are two versions (MUFFINN-DNmax and MUFFINN-DNsum) based on different strategies (*Cho et al., 2016*), and we use both versions in the comparison study. The interactome information of all the investigated methods is the iRefIndex gene interaction network from *Razick, Magklaras & Donaldson (2008)*. The pathway information used in DGPathinter is from KEGG, Reactome and BioCarta (*Ogata et al., 1999*; *Joshi-Tope et al., 2005*; *Nishimura, 2001*). In DGPathinter, the genes are ranked by their mutation scores from the matrix $G$. In the identification result of HotNet2, a higher delta score of a gene indicates a larger potential of being driver genes (details in Supplemental Information). For NBS, the genes are sorted according to their scores in the NBS profiles. In MUFFINN, the prediction scores of genes for MUFFINN-DNmax and MUFFINN-DNsum are used to prioritize the genes. To give a comprehensive view of the identification performance, we analyze all the investigated genes by precision–recall curves and average ranks of known driver genes over the prioritization results. For the top ranked genes, we further compare the fractions of known benchmarking genes among the results of these methods, and their related $p$ values of Fisher's exact test. Venn diagrams of the top ranked genes among the competing methods are also analyzed.

## Performance comparison

The overall performance of DGPathinter, HotNet2, NBS, MUFFINN-DNmax and MUFFINN-DNsum are illustrated as precision–recall curves in Fig. 2. When we use the known benchmarking cancer genes in NCG4.0 as a gold-standard, the precision–recall curves of DGPathinter are located over the other curves clearly for all the three types of cancers, indicating that DGPathinter yields the best identification performance among the four results on the datasets of the three types of cancers. Taking BRCA result as an example, the precisions of DGPathinter, HotNet2, NBS, MUFFINN-DNmax and MUFFINN-DNsum are 37.7%, 4.1%, 16.4%, 4.4% and 4.3% respectively when the recalls of the results are fixed at 5.0%. In the GBM results, the precisions of GBM-specific NCG4.0 genes are 37.6% for HotNet2, 45.8% for NBS, 0.8% for MUFFINN-DNmax and 3.6% for MUFFINN-DNsum when the recalls are 5.0%. In comparison, DGPathinter achieves a precision of 100.0% in the same situation. For the known experimental validated driver genes curated by CGC, we also draw the precision–recall curves of the four investigated results for the CGC gene list. In consistency with the NCG4.0 results,

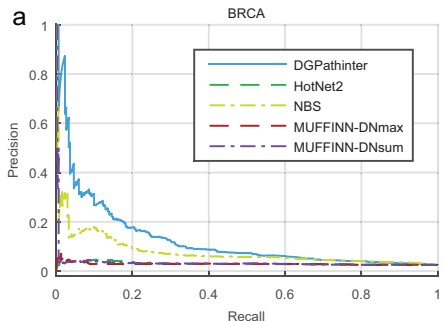 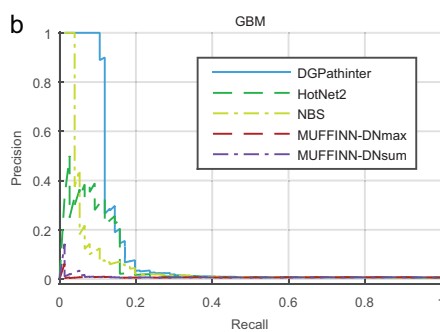 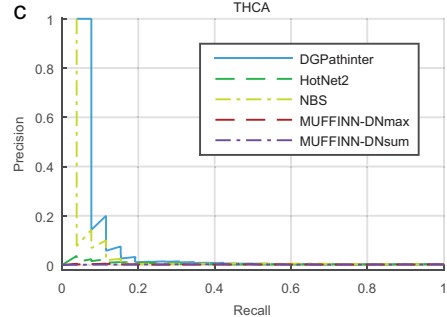

**Figure 2 Precision–recall curves of the prioritization results of the investigated methods for cancer specific known driver genes curated by NCG4.0 (*An et al., 2014*) on (a) BRCA, (b) GBM and (c) HNSC datasets, where blue, dark green, light green, dark red and violet lines represent the curves of DGPathinter, HotNet2, NBS, MUFFINN-DNmax and MUFFINN-DNsum, respectively.** Different points on a same curve represent the precisions and recalls at different thresholds of the results.

**Table 2 The average ranks of cancer specific known driver genes that are prioritized by the competing methods on BRCA, GBM and THCA dataset.**

| Known driver genes list | NCG4.0 | | | CGC | | |
|---|---|---|---|---|---|---|
| Method | BRCA | GBM | THCA | BRCA | GBM | THCA |
| DGPathinter | 67.5 | 28.1 | 15.1 | 12.2 | 7.7 | 15 |
| HotNet2 | 829.7 | 84.4 | 244.3 | 909.2 | 23.6 | 349.3 |
| NBS | 91.8 | 34.1 | 25.2 | 18 | 8.9 | 21.4 |
| MUFFINN-DNmax | 1731.4 | 1097.6 | 4940.1 | 1522.8 | 252.9 | 1663.2 |
| MUFFINN-DNsum | 1271.9 | 920.6 | 5666.6 | 1918.6 | 103.2 | 3642.5 |
| Random | 6064.5 | 6064.5 | 6064.5 | 6064.5 | 6064.5 | 6064.5 |

**Note:**
The evaluation cancer specific known driver genes are from NCG4.0 (*An et al., 2014*) (left part of table) and CGC (*Futreal et al., 2004*) (right part of table).

a similar phenomenon can also be observed that the identification performance of our approach outperforms the results of the other competing methods (Fig. S4). For example, DGPathinter gives a precision of 77.5% on the BRCA data and 100.0% on the GBM data when recalls are at 10.0%, which are also higher than those of the other competing methods in the same situation. To assess whether and to which extent the difference between the performance of our method and previous approaches is statistically significant, we apply the non-parametric Friedman test on the areas under the curve (AUCs) of precision–recall curves among the three investigated cancers (Table S1). The AUCs for known NCG4.0 and CGC genes yields $p$ values of 0.03 and 0.02, respectively, indicating that the difference between the performance of the investigated methods is statistically significant.

We also evaluate average rank of the known cancer genes predicted by the investigated methods. For the NCG4.0 gene list, DGPathinter yields average rank of 67.5 for known breast cancer specific genes, which is smaller than the those of 829.7 for HotNet2, 91.8 for NBS, 1731.4 for MUFFINN-DNmax, 1271.9 for MUFFINN-DNsum (Table 2) and 6064.5 for random selection. This result demonstrates that the benchmarking cancer genes in the result of our approaches are ranked much closer to the top in average, when compared

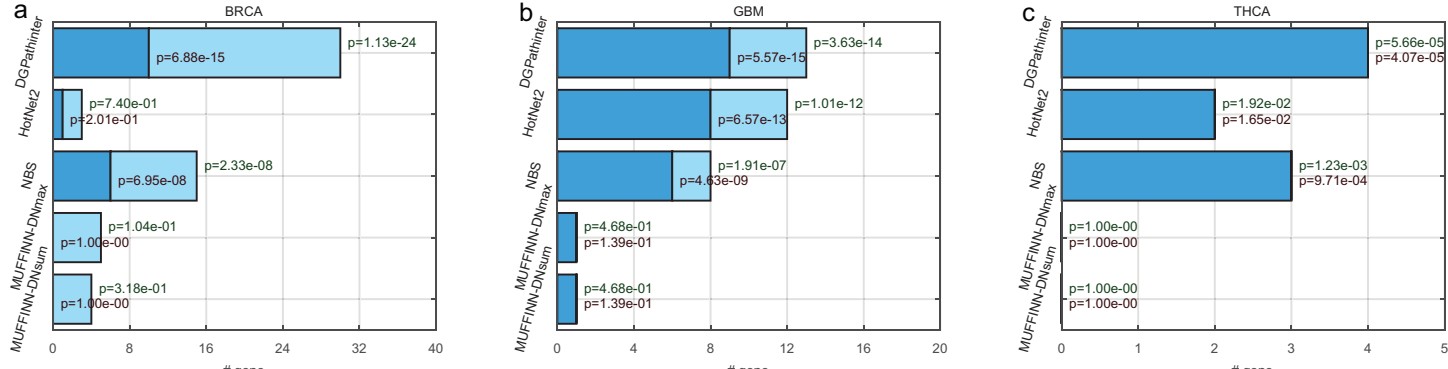

**Figure 3** Bar plot of numbers of known cancer specific driver genes that are selected in the top 100 genes among the competing prioritization results, for (a) BRCA, (b) GBM and (c) THCA respectively. The dark blue bars represent the number of CGC genes (*Futreal et al., 2004*), and the light blue bars represent the number of statistically inferred candidates genes in NCG4.0 (including both CGC genes and statistically inferred genes) (*An et al., 2014*). The dark red texts at the top of the dark blue bars indicate the *p* values of Fisher's exact test on the selected genes for cancer specific CGC gene, while the dark green texts at the top of the light blue bars represent the *p* values for cancer-specific NCG4.0 genes.

with those of other three results. When we examine the CGC experimentally validated genes, our approach also yields the smallest average rank among the competing methods (Table 2). The average ranks for breast cancer specific CGC gene list are 67.5, 829.7, 91.8, 1731.4, 1271.9 and 6064.5 for DGPathinter, HotNet2, NBS, MUFFINN-DNmax, MUFFINN-DNsum and random selection, respectively. We also apply the non-parametric Friedman test on the average ranks of known NCG4.0 and CGC genes in the detection results of the competing methods, yielding *p* values of 0.02 and 0.02 respectively. The aforementioned investigations suggest that DGPathinter shows a promising capability for prioritizing known cancer genes than those of the other competing approaches.

## Evaluation of top ranked genes

For the top ranked genes, the top 100 genes of the five prioritization results are selected for further evaluation. In BRCA result, the top 100 genes for HotNet2, NBS, MUFFINN-DNmax and MUFFINN-DNsum include 3, 15, 5 and 4 genes in NCG4.0 list respectively. In contrast to the top 100 genes for DGPathinter, there are 30 genes matched in the NCG4.0 benchmarking genes (Fig. 3). The *p* value of Fisher's exact tests on the result of the DGPathinter on BRCA is 1.13–24, indicating that the selected NCG4.0 genes are significantly enriched for NCG4.0 genes. For the GBM results, DGPathinter, HotNet2, NBS, MUFFINN-DNmax and MUFFINN-DNsum identify 13, 12, 8, 1 and 1 NCG4.0 genes, with related *p* values of 3.63e–14, 1.01e–12, 1.91e–07, 4.68e–01 and 4.68e–01 respectively. Compared with the *p* values of the other identification results, DGPathinter yields the smallest *p* values among the competing results (Fig. 3). These results demonstrate that DGPathinter performs the best among these methods in detecting NCG4.0 benchmarking cancer genes. Furthermore, we investigate the numbers of selected top 100 genes that are also CGC experimentally validated driver genes. For BRCA data, there are 10, 1, 6 CGC driver genes detected by DGPathinter, HotNet2 and NBS, with

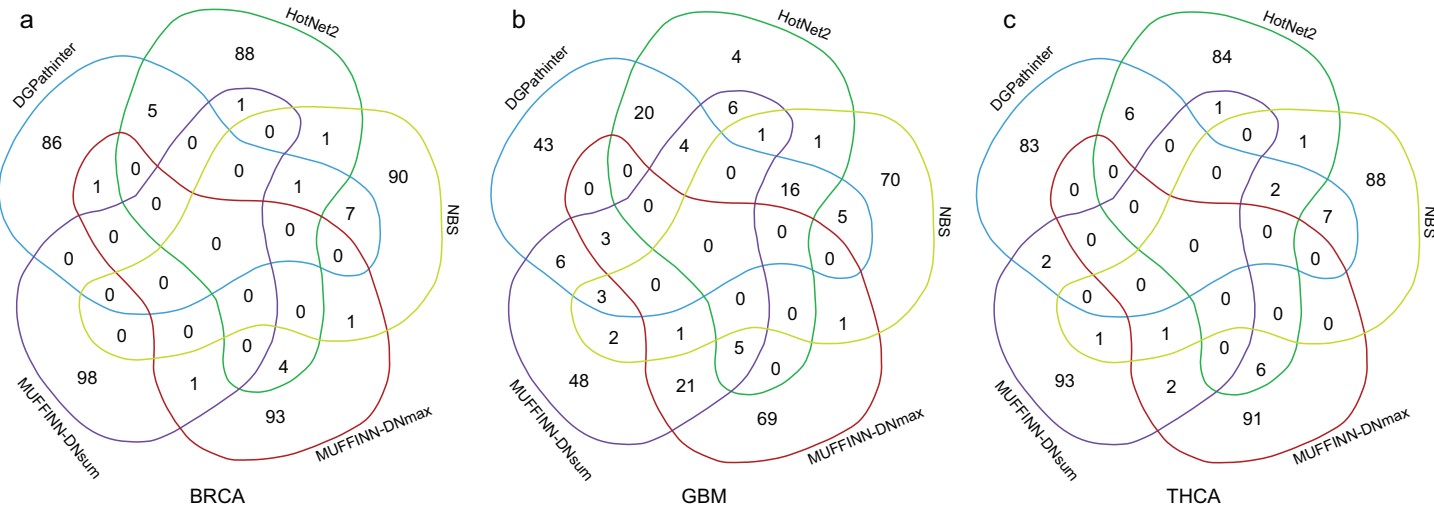

**Figure 4 Venn diagrams of the top 100 genes in the results of DGPathinter (blue circle), HotNet2 (dark green circle) and NBS (light green circle), MUFFINN-DNmax (dark red circle) and MUFFINN-DNmax (violet circle) on (a) BRCA, (b) GBM and (c) THCA datasets.**

*p* values of 6.88e−15, 2.01e−01 and 6.95e−08 respectively. For GBM datasets, there are nine CGC genes captured by DGPathinter, of which the Fisher's exact test *p* value is 5.57e−15 and is smaller than those of other investigated methods (eight CGC genes for HotNet2 with *p* value of 6.57e−13, six CGC genes for NBS with *p* value of 4.43−e09 and one CGC gene for both MUFFINN-DNmax and MUFFINN-DNsum with *p* values of 1.39e−01). For THCA-specific driver genes detected by the competing methods, the NCG4.0 genes completely overlap the CGC genes, and it can be observed that DGPathinter achieves the best performance among the competing methods. When we apply the Friedman test on the proportions of known NCG4.0 and CGC genes in the top 100 genes detected by DGPathinter on the investigated cancers, we obtain *p* values of 0.05 and 0.02 respectively. We further change the number of top-ranked genes considered to see how the changes affect the results of the statistical test. For the proportions of known NCG4.0 and CGC genes in the top 200 genes of the results of the competing methods, the *p* values of Friedman test are 0.04 and 0.02 respectively; for the top 300 genes, the *p* values are 0.03 for known NCG4.0 genes and 0.02 for known CGC genes. These results demonstrate that there is a statistically significant difference between the performance of the competing methods.

Furthermore, we compare the top 100 genes among the five prioritization results and draw Venn diagram of their results for three types of cancers respectively (Fig. 4). For BRCA dataset, 14 genes identified by DGPathinter are also detected by at least of one of the other results. For example, GATA3 gene is identified by DGPathinter, HotNet2 and NBS. As reported in a previous study (*Usary et al., 2004*), variants in GATA3 gene may have contribution to tumorigenesis in ESR1-positive breast cancers. Another study also shows that GATA3 mutations have the potential to be associated with aberrant nuclear localization, reduced transactivation and cell invasiveness in breast cancers (*Gaynor et al., 2013*). For GBM dataset, there are totally 16 genes shared in the detection

results of DGPathinter, HotNet2 and NBS, including CGC curated GBM specific driver genes TP53, PTEN, PIK3R1, EGFR and PIK3CA (*Futreal et al., 2004*), and NCG4.0 inferred GBM specific driver candidates ERBB2 and RB1 (*An et al., 2014*). For THCA dataset, HRAS is co-detected by DGPathinter, HotNet2 and NBS. Although HRAS is not a THCA specific driver gene curated by neither CGC nor NCG4.0, it is reported as driver gene for infrequent sarcomas and some other rare tumor types by CGC (*Futreal et al., 2004*).

Moreover, some genes detected by DGPathinter are also captured by MUFFINN-DNmax or MUFFINN-DNsum. Taking GBM results as an example, MDM4 gene is shared by the results of DGPathinter, MUFFINN-DNmax and MUFFINN-DNsum, which is reported to be the driver gene in bladder cancer, glioblastoma and retinoblastoma by CGC (*Futreal et al., 2004*). The KRAS gene is identified by DGPathinter, HotNet2 and MUFFINN-DNsum, which is also a driver gene in several types of cancers reported by CGC (*Futreal et al., 2004*). The FLT4 gene is included in the results of DGPathinter, NBS and MUFFINN-DNsum, and is curated as a driver gene of soft tissue sarcoma by CGC (*Futreal et al., 2004*). In addition to the genes shared by DGPathinter and other methods, there are also some genes unique to the results of DGPathinter. For example, TSH gene is a breast cancer driver gene curated by CGC, which is only detected by DGPathinter on the BRCA dataset. A number of CGC curated GBM specific driver genes, including AKT1, CDH1, MAP2K4, NCOR1 and TBX3 (*Futreal et al., 2004*), are also unique to the results of DGPathinter on the GBM dataset. For the results of THCA dataset, the CGC-curated THCA specific driver gene TERT is only identified by DGPathinter but not the other competing methods (*Futreal et al., 2004*). The full lists of the top 100 genes detected by DGPathinter on BRCA, GBM and TCHA, along with the methods that co-detect them, are demonstrated in Tables S2–S4, respectively.

## Pathway analysis

In addition to driver gene identification, DGPathinter can also provide highly scored pathways during the driver gene detection processing. We further analyze the top 30 scored pathways in the results of DGPathinter, and find some well-known cancer related pathways such as P53 pathway, PTEN pathway, P38MAPK events pathway, ATM pathway (Table 3). In the results of the BRCA dataset, the top one pathway is the GATA3 pathway curated by the BIOCARTA database, which is reported to be highly associated with breast cancer. For example, the GATA3 pathway is reported to play an important role in reducing E-cadherin in breast cancer tissues (*Tu et al., 2017*). Meanwhile, the top ranked pathway in the GBM results is the RB pathway curated by BIOCARTA, which is also found in the BRCA results. Reported by previous studies (*Chow et al., 2011*; *Sherr & McCormick, 2002*), mutated RB1 pathway is one of the obligate events in the pathogenesis of glioblastomas. Especially, thyroid cancer pathway is found in the results of DGPathinter on THCA dataset, and glioma pathway is in the results on the GBM dataset. Some other cancer-related pathways are also included in the lists of top ranked pathways, such as the GAB1 signalosome pathway, the signaling to RAS, the

**Table 3 Top 30 scored pathways in the results of DGPathinter on somatic mutation datasets of BRCA, GBM and THCA.**

| Rank | BRCA | GBM | THCA |
|---|---|---|---|
| 1 | BIOCARTA GATA3 PATHWAY | BIOCARTA RB PATHWAY | REACTOME SHC MEDIATED SIGNALLING |
| 2 | BIOCARTA RNA PATHWAY | BIOCARTA RNA PATHWAY | REACTOME SOS MEDIATED SIGNALLING |
| 3 | REACTOME GAB1 SIGNALOSOME | BIOCARTA ARF PATHWAY | REACTOME P38MAPK EVENTS |
| 4 | BIOCARTA ARF PATHWAY | BIOCARTA TEL PATHWAY | REACTOME GRB2 EVENTS IN EGFR SIGNALING |
| 5 | BIOCARTA TRKA PATHWAY | BIOCARTA P53 PATHWAY | REACTOME SIGNALLING TO P38 VIA RIT AND RIN |
| 6 | BIOCARTA HCMV PATHWAY | BIOCARTA CTCF PATHWAY | REACTOME SHC RELATED EVENTS |
| 7 | BIOCARTA RB PATHWAY | BIOCARTA PML PATHWAY | BIOCARTA VITCB PATHWAY |
| 8 | BIOCARTA LONGEVITY PATHWAY | BIOCARTA TID PATHWAY | REACTOME FRS2 MEDIATED ACTIVATION |
| 9 | BIOCARTA CTCF PATHWAY | BIOCARTA PTEN PATHWAY | REACTOME PURINE RIBONUCLEOSIDE MONOPHOSPHATE BIOSYNTHESIS |
| 10 | BIOCARTA ACH PATHWAY | REACTOME SEMA4D INDUCED CELL MIGRATION AND GROWTH CONE COLLAPSE | BIOCARTA IL3 PATHWAY |
| 11 | BIOCARTA GCR PATHWAY | BIOCARTA P53HYPOXIA PATHWAY | BIOCARTA ACE2 PATHWAY |
| 12 | BIOCARTA GLEEVEC PATHWAY | BIOCARTA ATRBRCA PATHWAY | REACTOME TIE2 SIGNALING |
| 13 | BIOCARTA BCELLSURVIVAL PATHWAY | BIOCARTA IGF1MTOR PATHWAY | BIOCARTA PLATELETAPP PATHWAY |
| 14 | BIOCARTA CDC42RAC PATHWAY | REACTOME GAB1 SIGNALOSOME | REACTOME SIGNALLING TO RAS |
| 15 | BIOCARTA P53 PATHWAY | BIOCARTA ATM PATHWAY | KEGG ETHER LIPID METABOLISM |
| 16 | BIOCARTA IL7 PATHWAY | BIOCARTA G1 PATHWAY | KEGG THYROID CANCER |
| 17 | BIOCARTA RAC1 PATHWAY | REACTOME SEMA4D IN SEMAPHORIN SIGNALING | BIOCARTA AMI PATHWAY |
| 18 | BIOCARTA ERK5 PATHWAY | BIOCARTA G2 PATHWAY | REACTOME SIGNALLING TO ERKS |
| 19 | BIOCARTA CTLA4 PATHWAY | BIOCARTA CHEMICAL PATHWAY | BIOCARTA INTRINSIC PATHWAY |
| 20 | BIOCARTA PTEN PATHWAY | KEGG ENDOMETRIAL CANCER | KEGG PENTOSE PHOSPHATE PATHWAY |
| 21 | BIOCARTA PML PATHWAY | BIOCARTA MTOR PATHWAY | REACTOME DOWN STREAM SIGNAL TRANSDUCTION |
| 22 | BIOCARTA NGF PATHWAY | KEGG BLADDER CANCER | REACTOME PURINE METABOLISM |
| 23 | REACTOME TIE2 SIGNALING | BIOCARTA EIF4 PATHWAY | BIOCARTA BAD PATHWAY |
| 24 | BIOCARTA ATM PATHWAY | KEGG NON SMALL CELL LUNG CANCER | KEGG GLYCEROPHOSPHOLIPID METABOLISM |
| 25 | BIOCARTA ATRBRCA PATHWAY | KEGG GLIOMA | KEGG BLADDER CANCER |
| 26 | BIOCARTA TEL PATHWAY | KEGG MELANOMA | KEGG TYROSINE METABOLISM |
| 27 | BIOCARTA TID PATHWAY | KEGG PANCREATIC CANCER | KEGG ALANINE ASPARTATE AND GLUTAMATE METABOLISM |
| 28 | BIOCARTA IGF1MTOR PATHWAY | KEGG PROSTATE CANCER | KEGG MTOR SIGNALING PATHWAY |
| 29 | REACTOME CD28 DEPENDENT PI3K AKT SIGNALING | BIOCARTA MET PATHWAY | KEGG ENDOMETRIAL CANCER |
| 30 | REACTOME FURTHER PLATELET RELEASATE | REACTOME PI3K AKT SIGNALLING | REACTOME SIGNALING BY EGFR |

MTOR signaling pathway, the non-small cell lung cancer pathway, the melanoma pathway, the pancreatic cancer pathway, the prostate cancer pathway, the bladder cancer pathway and the endometrial cancer pathway.

## DISCUSSION

In this paper, we propose a knowledge-driven matrix factorization framework called DGPathinter to identify driver genes from mutation data with prior knowledge from interactome and pathways incorporated. The knowledge of pathways is incorporated by maximizing the correlation between the pathway scores and their relations of mutation scores (*Chen & Zhang, 2016*). Meanwhile, the knowledge of interactome is utilized by graph Laplacian regularization with the gene interaction network. To integrate the information from pathways, interactome and mutation data, matrix factorization framework is adopted, which can also help addressing the problem of tumor sample heterogeneity. When comparing DGPathinter with three existing methods on three TCGA cancer mutation datasets (BRCA, GBM and THCA), we observe that DGPathinter achieves better performance than the other competing methods on precision–recall curves. The average ranks of known driver genes prioritized by DGPathinter are also smaller than those of the other existing methods. The top ranked genes detected by DGPathinter are highly enriched for the known driver genes when analyzed by Fisher's exact test, and the *p* values for DGPathinter are also more significant than those of the other investigated methods. While some known driver genes are shared in the detection results of DGPathinter and the existing methods, DGPathinter also identifies some known driver genes that are not detected by the other investigated methods. In addition, most of the top ranked scored pathways in the results of DGPathinter are cancer progression-associated pathways.

The promising performance of DGPathinter in the identification of driver genes may be due to three potential reasons. First, prior knowledge from pathways is important for understanding the roles of genes in tumors (*Parsons et al., 2008*; *Cancer Genome Atlas Research Network, 2008*; *Vaske et al., 2010*), and incorporating information from pathways has the potential of promoting the detection power of driver genes. Second, since the cooperatively dysregulated genes are correlated with cancer formation and progression (*Vandin, Upfal & Raphael, 2011*; *Leiserson et al., 2014*; *Hofree et al., 2013*; *Cho et al., 2016*), gene interaction network information from interactome can help in determining the influence of somatic mutations between the interacted genes. Third, the sample heterogeneity issue that driver genes may mutate in different samples is reported as a confounding factor in driver gene identification (*Cancer Genome Atlas Network, 2012*), and matrix factorization framework is capable of analyzing heterogeneous samples (*Lee et al., 2010*; *Sill et al., 2011*; *Xi & Li, 2016*). To investigate the individual contribution of network information or pathway information on the performance of our method, we calculate the results with only the network information, the results with only the pathway information and the results with no prior information (i.e., matrix factorization) by removing the two terms of pathway information, the term of network regularization and all the three terms of prior information respectively. Through the evaluation results in Figs. S5 and S6, we can see that the results with prior information from both network and pathways achieve better performance than the results with only network information, the results with only pathway information, and the results with no

 

prior information. These comparison results indicate that the detection results of our methods are contributed by the prior knowledge coming from networks and pathways.

Note that the previous network-based methods can incorporate pathway information when they use network that contains interaction data derived from pathway databases, such as public molecular interaction database STRING. In comparison, iRefIndex is an interaction database that only assembles data from other primary interaction databases, but does not provide the coverage of pathway interactions that is offered by STRING. Therefore, we compare our method against the previous network-based approaches by using STRING instead of iRefIndex. The comparison results with the usage of STRING are evaluated by precision–recall curves (Fig. S7 for CGC and S8 for NCG4.0), the average ranks of known cancer genes (Table S5 for CGC and NCG4.0) and the proportions of known cancer genes in the top 100 genes with $p$ values of Fisher's exact test (Fig. S9 for CGC and NCG4.0). We observe that the performance of HotNet2 and MUFFINN-DNsum with STRING are increased when compared with their performance with iRefindex. This phenomenon may be due to the fact that iRefIndex does not provide the coverage of pathway interactions that is offered by STRING. For DGPathinter, the detection results still outperform those of the other network-based methods when STRING is used. When we further apply the statistical validation on the detection results of these competing methods with STRING, the validation results of the Friedman test demonstrate that the difference between the detection results of the investigated methods is statistically significant (details in Supplemental information). Consequently, our approach provides an added value over previous network-based approaches through the use of information on pathway boundaries, which is not explicitly included in STRING.

Despite the achievement, there are also some questions for further investigation. Since there seems to be a bias among network-based driver gene identification methods where hot nodes and their neighbors are always identified as candidate drivers, we further investigate how many of the top 100 DGPathinter-output genes are neighbors of TP53. For BRCA, GBM and THCA, the numbers of the top 100 genes detected by DGPathinter that are also neighbors of TP53 are 8, 18 and 9, respectively, which is much less than 427 (the number of neighbors of TP53 in iRefIndex). Accordingly, it seems that the results of DGPathinter are less affected by the bias among network-based driver gene identification methods. Furthermore, there is a possibility that some genes may contain both driver and passenger mutations, and this problem is not addressed in the experimental design in our work. The current approach focuses on gene-level predictions, and it cannot yet make predictions at the level of individual mutations. In Fig. S5, using network information does not change the performance for the datasets of the three types of cancers. When we further investigate this phenomenon, we find that some non-benchmarking genes included in top ranked genes in the result with network information are different than those in the result with no prior information, although the known benchmarking genes included in the two results are the same. In addition, a possible expansion to DGPathinter would be to integrate multi-omic data from not only mutations but also from copy number alternation, gene expression and DNA methylation of genes, which also play

important roles in activating oncogenes and inactivating tumor suppressors (*Yang et al., 2017*). Another interesting topic for future work is to generalize the framework of DGPathinter to pan-cancer analysis, in which the samples of numerous different cancer types is combined as one large dataset and some driver genes across many types of cancers will be identified in this case (*Leiserson et al., 2014*). In conclusion, DGPathinter is an efficient method for prioritizing driver genes, and yields a sophisticated perspective of cancer genome by utilizing prior knowledge from interactome and pathways.

## ACKNOWLEDGEMENTS

We would like to thank Changran Zhang and the anonymous reviewers for insight and help with revisions of this manuscript.

### Funding

This work was supported by the National Natural Science Foundation of China (Nos. 61571414, 61471331 and 31100955). There was no additional external funding received for this study. The funders had no role in study design, data collection and analysis, decision to publish, or preparation of the manuscript.

### Grant Disclosures

The following grant information was disclosed by the authors:
National Natural Science Foundation of China: 61571414, 61471331 and 31100955.

### Competing Interests

The authors declare that they have no competing interests.

### Author Contributions

- Jianing Xi conceived and designed the experiments, performed the experiments, analyzed the data, contributed reagents/materials/analysis tools, wrote the paper, prepared figures and/or tables, performed the computation work and reviewed drafts of the paper.
- Minghui Wang conceived and designed the experiments, analyzed the data and reviewed drafts of the paper.
- Ao Li wrote the paper, reviewed drafts of the paper.

### Data Availability

  GitHub: https://github.com/USTC-HIlab/DGPathinter

### Supplemental Information

Supplemental information for this article can be found online at http://dx.doi.org/10.7717/peerj-cs.133#supplemental-information.

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
