# Peer review of "DGPathinter: a novel model for identifying driver genes via knowledge-driven matrix factorization with prior knowledge from interactome and pathways"

_PeerJ Computer Science, doi:10.7717/peerj-cs.133_

## Round 0.1 · original submission · Major Revisions

Please thoroughly address all points raised by the reviewers, as they have raised several concerns about justification of methods, contributions to the different parts of the domain knowledge used by the method, statistical validation of results or clarifying the differences from your previous work in Scientific Reports.

Reviewer 1 ·

Basic reporting

The article has a clear and logical structure, but I noticed several typos and grammatical errors in the manuscript, which should be corrected. Attached below, I provide a list with some of these errors and suggestions for corrections, but this is not comprehensive. I therefore strongly suggest to revise the language of the manuscript with the help of a native speaker.
Figure 1 should ideally also outline the meaning of the matrices F and V, and how they are integrated into the analysis workflow, since they contain key information for the proposed approach. I have no comments on the other figures and tables.
Finally, the authors should also discuss how their method relates to their previous approach published in Scientific Reports (Vol. 7, Article number: 2855, 2017, doi:10.1038/s41598-017-03141-w).

List of grammar/spelling errors and suggested corrections:
- Abstract, line 19: "prior knowledge from interactome" --> "prior knowledge from an interactome";
- Line 36: "cataloging driver genes from genes with passenger mutations" --> "distinguishing driver genes from genes with passenger mutations";
- Line 49: "another research on driver genes" --> "another research study on driver genes";
- Line 51: "Bayesian approach is also applied" --> "Bayesian approaches are also applied";
- Line 63: "their interacted neighbors" --> "their interacting neighbors";
- Line 75: "Instead of network propagation, MUFFINN prioritizes" --> "Instead of using network propagation, MUFFINN prioritizes";
- Line 76: "their directed neighbors" --> "their direct neighbors";
- Line 100: "use graph Laplacian technique" --> "use a graph Laplacian technique";
- Line 100: "from interaction network" --> "from an interaction network"; line 123: "in the related gene for which the related tumor sample," --> "in the respective gene and tumor sample.

Experimental design

Most aspects of the experimental design are clear and appropriate; however, I see a potential issue resulting from the authors' claim that previous methods have only used network information but no pathway information (line 21-22): Public molecular interaction databases like STRING, which have been used for previous cancer driver gene mutation prediction approaches (e.g. the approach cited in line 68), also contain interaction data derived from pathway databases (in fact, this is one of the main data sources for STRING). It is conceivable that the authors' approach may still provide an added value over previous network-based approaches through the use of information on pathway boundaries, which is not explicitly included in STRING, but to test this, the authors would need to compare their method against a network-based approach using STRING data (or another network data source exploiting pathway information) rather than an approach based on iRefIndex (an interaction database that only assembles data from other primary interaction databases, but does not provide the coverage of pathway interactions that is offered by STRING).
A question that is not discussed in the manuscript and not addressed in the experimental design is whether some genes may contain both driver and passenger mutations - while the authors' current approach focuses on gene-level predictions, can it also help to make predictions at the level of individual mutations? This should also be discussed in the manuscript.

Validity of the findings

The authors provide multiple sources of quantitative evidence to support the claim that their cancer driver gene prediction method is superior to previous approaches. However, while the presented precision-recall curves suggest a clear difference between the performance of the authors' method and previous approaches, the authors should additionally use a statistical test to assess whether and to which extent the difference is also statistically significant. For example, the authors could compute the areas under the curve and compare them with the non-parametric Friedman test.
The same applies to the comparison of average ranks and the evaluation of top-ranked genes, which should be accompanied by a statistical test (the authors should also briefly comment on whether changes in the number of top-ranked genes considered affect the results).
Importantly, the potential issue regarding the use of networks without interaction data from pathway databases for the comparative evaluation, mentioned in the experimental design section, may also affect the validity of the findings, and the validation results should therefore be recomputed for a network data source like STRING that contains pathway-derived interaction data.

Additional comments

As a minor point, in line 267 the authors probably wanted to refer to the "smallest average rank" instead of the "largest average rank" (in Table 2 lower average ranks represent better results, therefore please check this again).

Reviewer 2 ·

Basic reporting

1) Prior literature on driver gene identification is well explained; however, more intuitive explanation is needed on why matrix factorization is an appropriate model for this task.

2) There's typo in Table 2 where the proposed method is incorrectly named as 'PathInterDG'

3) There's typo in Discussion: 'which are also play important'

Experimental design

1) Based on the presented results, the proposed method seems to be better in identification of known cancer driver genes. However, it would be useful to see the individual contribution of network information or pathway information; as currently it is not clear whether this success is due to the model (i.e., matrix factorization) or due to the prior knowledge coming from networks and pathways.

2) The authors state that "λC, λV and λL are empirically set to 0.01, 0.01 and 0.1", how stable is the model with respect to these parameters. Namely, do the results vary a lot if these parameters are changed?

Validity of the findings

1) In "Driver gene identification" section, the authors state that "In the identification result of HotNet2, a higher delta score of a gene indicates a larger potential of being driver genes". Here, it is not clear what delta score represents. Also, to the best of my knowledge, it is not possible to rank the genes in any way in Hotnet2 results. In Hotnet2, the edges with weights smaller than a threshold is used, and all the genes that are in strongly connected components in the remaining graph are identified as candidate driver genes. As such, the authors should better explain how they rank genes in Hotnet2 results.

2) How many of the DGPathInter-output genes (e.g. top 100) are neighbors of TP53? It would be useful to mention this. There seems to be a bias among network-based driver gene identification methods where hot nodes and their neighbors are always identified as candidate drivers.

---

## Round 0.2 · Minor Revisions

Please address the remaining minor concerns of reviewer 2.

Reviewer 1 ·

Basic reporting

no comment

Experimental design

no comment

Validity of the findings

no comment

Additional comments

The authors have addressed my comments and their revisions have improved the manuscript.

Reviewer 2 ·

Basic reporting

Most of the problems with English are now fixed but the manuscript still has some typos and errors:
p3: the revised computational model -> a revised computational model
p4: heterogeneity tumor samples (??) Do the authors mean heterogeneous?
p5: to integrated -> to integrate
Table 2: PathInterDG has to be replaced by DGPathinter again.
p13: performances of our method -> performance of our method --same mistake is done throughout the page

Experimental design

The additional experiments done by the authors have strengthened the claims of the paper. In Figure S5, using network information doesn't change the performance for all three cancer types. The authors should add some discussion about this result.

Validity of the findings

no comment

---

## Round 0.3 · Minor Revisions

Reviewer 2 was asking you to address the observation in figure S5 about the lack of improvement when using the network information. The reply "we will look at it in future work" is not good enough. Please, provide a more detailed answer.

---

## Round 0.4 · accepted · Accept

I am happy with the details provided by the authors in the latest revision. However, before final publication please have a native speaker proof-reading the manuscript, as there are still many grammar errors, especially in the introduced text.